# Learning Locality and Isotropy in Dialogue Modeling

**Han Wu**[1,2], **Haochen Tan**[1,2], **Mingjie Zhan**[3], **Gangming Zhao**[3], **Shaoqing Lu**[3],
**Ding Liang**[3], **Linqi Song**[1,2,*]
[1]Department of Computer Science, City University of Hong Kong
[2]City University of Hong Kong Shenzhen Research Institute
[3]Sensetime Research
`{hanwu32-c,haochetan-2}@my.cityu.edu.hk`
`linqi.song@cityu.edu.hk`

## Abstract

Existing dialogue modeling methods have achieved promising performance on various dialogue tasks with the aid of Transformer and the large-scale pre-trained language models. However, some recent studies revealed that the context representations produced by these methods suffer the problem of *anisotropy*. In this paper, we find that the generated representations are also not *conversational*, losing the conversation structure information during the context modeling stage. To this end, we identify two properties in dialogue modeling, i.e., locality and isotropy, and present a simple method for dialogue representation calibration, namely SimDRC, to build isotropic and conversational feature spaces. Experimental results show that our approach significantly outperforms current state-of-the-art models on three open-domain dialogue tasks with eight benchmarks. More in-depth analyses further confirm the effectiveness of our proposed approach. We release the code at `https://github.com/hahahawu/SimDRC`.

## 1 Introduction

*Dialogue modeling* (Serban et al., 2016; Mehri et al., 2019; Liu et al., 2021) is to encode the raw text of the input dialogue to the contextual representations. Although the Transformer-based dialogue modeling methods (Hosseini-Asl et al., 2020; Liu et al., 2021) have achieved great success on various dialogue tasks, there are still some impediments in these methods that are not well explored nowadays. Specifically, recent studies (Ethayarajh, 2019; Su et al., 2022) have revealed that on dialogue generation tasks, the representations produced by existing dialogue modeling methods are *anisotropic*, i.e. features occupy a narrow cone in the vector space, thus leading to the problem of degeneration. To alleviate this problem, previous solutions (e.g. SimCTG) (Su et al., 2021; 2022) encourage the model to learn isotropic token embeddings by pushing away the representations of distinct tokens. While building the more discriminative and isotropic feature space, these methods still ignore learning dialogue-specific features, such as inter-speaker correlations and conversational structure information, in the dialogue modeling stage. Therefore, a question is naturally raised - are the representations produced by existing dialogue modeling methods really conversational?

To answer this question, in Figure 1(a), we showcase the cosine similarity matrix of token representations produced by BART (Lewis et al., 2020) that is well trained on response generation task. First, we can easily observe the phenomenon of anisotropy from the heatmap where the similarities of distinct tokens are relatively high, over 0.5 for most token pairs. Then, Figure 1(b) illustrates the similarity heatmap of token representations produced by SimCTG where the color is faded on the whole, suggesting the problem of anisotropy is relaxed. However, another critical problem still remains, is that the representations of tokens in different utterances are nearby to each other, making the utterance indistinguishable on the token representations. It is undesirable that no conversational features can be captured from the token similarity matrix while the matrix is produced by a "dialogue modeling" method trained on the dialogue task using dialogue data. Ideally, we expect that the

---
[*] Corresponding author.

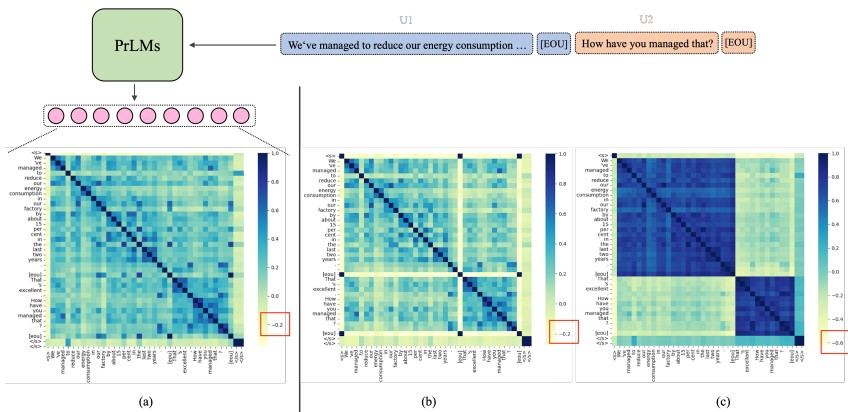

Figure 1: Illustrations of the token cosine similarity matrix produced by BART (a), SimCTG (Su et al., 2022) (b) and our proposed SimDRC (c).

representations of tokens within an utterance are close to voice a concentrated idea of the utterance, and the representations of different utterances are discriminative and isotropic to convey the maximal information of the dialogue. Accordingly, the ideal similarity matrix of token representations should be similar to Figure 1(c), where tokens within an utterance are intensive and different utterances are easily distinguishable on representations. Our motivation is that humans pay more attention to the central idea of the utterance rather than how the utterance is organized by words when humans utter, and humans also prefer to express more information with fewer utterances (Woolnough et al., 2021).

Based on the above observation and motivation, we identify two properties, i.e., locality and isotropy in dialogue modeling, and then present *SimDRC*, a **sim**ple **d**ialogue **r**epresentation **c**alibration method, which encourages the model to aggregate the representations of tokens within an utterance and push away the representations of distinct utterances. We evaluate our approach on three open-domain dialogue tasks, including multi-turn dialogue response generation (Li et al., 2017), conversational response retrieval (Lowe et al., 2015) and conversational semantic role labeling (Xu et al., 2021). The experimental results show that our approach achieves comparable or better performance against the current state-of-the-art methods across the three dialogue tasks on both automatic and human evaluations. In-depth analyses towards the effects of the hyper-parameters and the measurements of locality and isotropy further verify the effectiveness of our proposed approach.

## 2 RELATED WORK

### 2.1 DIALOGUE MODELING

Dialogue modeling is to transform the raw text of the dialogue to machine-readable representations, which is an indispensable step to most dialogue tasks (Li et al., 2017; Liu et al., 2021). To achieve this goal, conventional approaches (Serban et al., 2016; 2017; Xing et al., 2018) with recurrent neural networks (RNN) (Hochreiter & Schmidhuber, 1997; Mikolov et al., 2010) prefer to hierarchically learn the representations due to the long distance dependency problems in RNNs. With the remarkable success of Transformer (Vaswani et al., 2017) and pre-trained language models (Devlin et al., 2019; Raffel et al., 2020) on various NLP tasks, Transformer-based dialogue modeling methods (Hosseini-Asl et al., 2020; Gu et al., 2020; Liu et al., 2021; Wu et al., 2021) are widely used, and significantly outperform the traditional methods on many dialogue tasks, such as response generation (Li et al., 2017) and response retrieval (Lowe et al., 2015). In this work, we concentrate on the standard Transformer-based dialogue modeling method, which directly encodes the flattened dialogue context with the pre-trained language models. By studying the geometry of the representation space of the model, we find that the contextual representations produced by existing dialogue modeling methods are not isotropic and conversational.

### 2.2 REPRESENTATION CALIBRATION

Outside dialogue modeling, many other representation learning approaches also attempt to normalize their feature distributions from different perspectives. A bunch of studies theoretically verify that

uniformly distributing points on unit hypersphere can provide good representations (Bojanowski & Joulin, 2017; Wang & Isola, 2020). Specifically, Wang & Isola (2020) directly uses the alignment and uniformity properties to guide the representation learning, finally achieving comparable or even better performance on downstream tasks than the original contrastive learning framework. With the rapid spread of Transformer, a set of works (Dong et al., 2021; Su et al., 2022) also empirically explore the intrinsic properties of representations produced by the models, and find that the contextual words representations from most models are *anisotropy*, occupying a narrow cone in the feature space (Ethayarajh, 2019). This observation strongly agrees with our intuition that refining dialogue feature space is a desirable step. To this problem, SimCTG (Su et al., 2022) presents a token-level contrastive learning framework to calibrate the model's representation space regarding the anisotropy problem. However, the essential difference with our work is that we normalize the feature space on multi-granularity levels instead of just the token level, which matches better with the structural characteristics of the dialogue and produces more isotropic and conversational representations.

## 3 METHODOLOGY

### 3.1 DIALOGUE MODELING

Given a dialogue $D = \{u_1, u_2, ..., u_N\}$ consisting of $N$ utterances, where $u_i = \{w_{i,1}, w_{i,2}, ..., w_{i,|u_i|}\}$ is a sequence of words, the goal of dialogue modeling is to learn the contextual representation $\mathbf{H} = \{\mathbf{h}_{1,1}, ..., \mathbf{h}_{i,j}, ..., \mathbf{h}_{N,|u_N|}\}$ over the dialogue context. Typically, the Transformer-based dialogue modeling method obtain the dialogue contextual representations by taking the utterances as a consecutive sequence with special separate tokens, which is presented as:

$$\mathbf{H} = \text{PrLM}(u_1 \text{ [EOU] } u_2 \text{ [EOU] } ... u_n \text{ [EOU] [CONTEXT]}), \tag{1}$$

where [EOU] and [CONTEXT] are special tokens which would further be used to represent the whole utterance or dialogue. [EOU] and [CONTEXT] are respectively inserted as the last token of each utterance and the last token of the whole dialogue.

### 3.2 LOCALITY AND ISOTROPY DISTANCE

Conventional wisdom says that a good dialogue contextual representation should be informative, discriminative and structurally sensitive (Gu et al., 2020; Wu et al., 2021; Su et al., 2022). Therefore, we identify two properties for good dialogue representations as following:

- *Locality*: tokens within an utterance are mapped to nearby features, thus concentrating on expressing a core idea of the utterance.
- *Isotropy*: utterance representations are roughly uniformly distributed in the feature space, thus preserving as much information of the whole dialogue as possible and enabling the utterance discriminative to each other.

For locality, it encourages the learned feature representations of the tokens within an utterance to be similar. In practice, the special token (i.e., [EOU]) is often served as the representative token of the utterance to speak for the utterance (Zhu et al., 2020; Liu et al., 2021). Therefore, we model locality by bridging the gap between each token and the representative token of the utterance it belongs. The benefits of this design are 1) the representations of the tokens within an utterance are pushed close to the representation of the utterance representative token for better expressing the utterance's concentrated idea; 2) the representative token are encouraged to combine the information from all tokens within the utterance, thus becoming more informative and representative. Formally, given a dialogue $D = \{w_{1,1}, ..., w_{i,j}, ..., w_{N,|u_N|}\}$, for each token $w_{i,j(j \neq |u_i|)}$ in the context, we obtain the locality value by calculating the cosine similarity between the token and the corresponding utterance representative token $w_{i,|u_i|}$,

$$\text{locality\_value}_{(i,j)} = \frac{\mathbf{h}_{i,j}^T \mathbf{h}_{i,|u_i|}}{\|\mathbf{h}_{i,j}\| \cdot \|\mathbf{h}_{i,|u_i|}\|}, \tag{2}$$

where $\mathbf{h}_{i,j}$ is the representation of token $w_{i,j}$. The value will be larger when the two vectors are closer. Then, we compute the overall locality distance for the whole dialogue context as:

$$\text{locality\_distance}_{(D)} = \frac{1}{|u_1| + ... + |u_N| - N} \sum_{i=1}^{N} \sum_{j=1}^{|u_i|-1} \text{locality\_value}_{(i,j)}. \tag{3}$$

For isotropy, it encourages the model to push away the features of distinct utterances, thus preserving the maximal information of the dialogue and enabling the utterance discriminative and isotropic. Similar to locality, we model isotropy by enlarging the gap between distinct representative tokens. Formally, given two distinct utterances $u_i$ and $u_j$, we obtain the isotropy value by computing the cosine similarity between two utterance representative tokens,

$$\text{isotropy\_value}_{(i,j)} = \frac{\mathbf{h}_{i,|u_i|}^T \mathbf{h}_{j,|u_j|}}{\|\mathbf{h}_{i,|u_i|}\| \cdot \|\mathbf{h}_{j,|u_j|}\|}. \tag{4}$$

Similarly, the overall isotropy distance of the whole dialogue context is formulated as:

$$\text{isotropy\_distance}_{(D)} = \frac{1}{N \times (N-1)} \sum_{i=1}^{N} \sum_{j=1,i \neq j}^{N} \text{isotropy\_value}_{(i,j)}. \tag{5}$$

### 3.3 ADJUSTING FEATURE SPACE

Based on the locality distance and isotropy distance, we present locality loss, isotropy loss and SimDRC loss to adjust the feature space for modeling better representations.

**Locality loss**  attempts to close the representations of the tokens within an utterance to the representations of the corresponding utterance representative token. Therefore, we introduce locality loss, aiming at maximizing the locality distance defined in Equation 3,

$$\mathcal{L}_{\text{locality}} = \frac{1}{|u_1| + ... + |u_N| - N} \sum_{i=1}^{N} \sum_{j=1}^{|u_i|-1} \max(0, \delta_1 - \text{locality\_value}_{(i,j)}) \tag{6}$$

where $\delta_1 \in [0, 1]$ is the margin value. When $\delta_1$ equals to 0, the loss degrades to 0.

**Isotropy loss**  tries to push away the representations of distinct utterances. To this end, we introduce isotropy loss to minimize the isotropy distance defined in Equation 5,

$$\mathcal{L}_{\text{isotropy}} = \frac{1}{N \times (N-1)} \sum_{i=1}^{N} \sum_{j=1,i \neq j}^{N} \max(0, \delta_2 + \text{isotropy\_value}_{(i,j)}) \tag{7}$$

where $\delta_2 \in [-1, 1]$ is the margin value. When $\delta_2$ equals to -1, the loss degrades to 0.

**SimDRC loss**  combines the locality loss and isotropy loss to calibrate the representation space. Formally, we present the SimDRC loss as,

$$\mathcal{L}_{\text{SimDRC}} = \alpha \mathcal{L}_{\text{locality}}(\delta) + (1 - \alpha) \mathcal{L}_{\text{isotropy}}(\delta) \tag{8}$$

where $\delta \in [-1, 1]$ is the margin value, shared between the locality loss and isotropy loss. $\alpha$ is the hyper-parameter of loss weight. The SimDRC loss degenerates to locality loss when $\alpha = 1$, and to isotropy loss when $-1 < \delta \leq 0$ or $\alpha = 0$, and to 0 when $\delta = -1$. It is worth noting that, the isotropy loss is more sparse because it only works on a few utterance representative tokens while the locality loss computes on all tokens within the utterances. Therefore, the model would be biased to modeling

| Model | Method | DailyDialog | | | | | | LCCC | | | | | |
|-------|--------|-------------|--|--|--|--|--|------|--|--|--|--|--|
| | | BERTScore | | | BAS↑ | BT↑ | Dis2/4↑ | BERTScore | | | BAS↑ | BT↑ | Dis2/4↑ |
| | | P↑ | R↑ | F↑ | | | | P↑ | R↑ | F↑ | | | |
| BART | greedy | 12.31 | 10.51 | 11.34 | -3.61 | 0.404 | 0.305/0.702 | 6.82 | 9.30 | 7.75 | -6.19 | 0.206 | 0.133/0.481 |
| | beam | 12.29 | 12.85 | 12.05 | -3.58 | 0.408 | 0.301/0.676 | 7.52 | 9.29 | 8.10 | -6.21 | 0.214 | 0.178/0.534 |
| | nucleus | 12.50 | 12.85 | 12.15 | -3.58 | 0.410 | 0.303/0.686 | 7.55 | 9.34 | 8.14 | -6.19 | 0.215 | 0.172/0.521 |
| | contrastive | 6.34 | 6.92 | 6.60 | -4.23 | 0.270 | 0.273/0.631 | 4.64 | 5.62 | 4.81 | -6.31 | 0.175 | 0.155/0.503 |
| SimCTG ($\rho$=0.5) | greedy | 11.39 | 9.69 | 10.01 | -3.60 | 0.280 | 0.280/0.642 | 6.82 | 9.30 | 7.75 | -6.21 | 0.216 | 0.178/0.533 |
| | beam | 10.70 | 11.96 | 10.85 | -4.23 | 0.270 | 0.237/0.613 | 7.48 | 9.24 | 8.06 | -6.31 | 0.175 | 0.155/0.503 |
| | nucleus | 10.85 | 12.56 | 10.96 | -3.62 | 0.395 | 0.289/0.672 | 6.82 | 9.31 | 7.75 | -6.18 | 0.206 | 0.133/0.481 |
| | contrastive | 11.35 | 12.77 | 11.45 | -3.60 | 0.408 | 0.283/0.650 | 7.66 | 9.42 | 8.23 | -6.19 | 0.218 | 0.173/0.524 |
| SimDRC ($\delta$=0.7, $\alpha$=0.3) | greedy | 12.43 | 12.91 | 12.13 | -4.09 | 0.288 | 0.275/0.644 | 6.92 | 9.34 | 7.82 | -6.36 | 0.175 | 0.154/0.474 |
| | beam | 12.67 | 14.12 | 12.89 | -3.59 | 0.408 | 0.285/0.651 | 7.65 | 9.33 | 8.19 | -6.19 | 0.216 | 0.173/0.524 |
| | nucleus | **12.85** | **14.14** | **12.97** | **-3.55** | **0.419** | 0.318/0.711 | **7.77** | **9.48** | **8.32** | -6.21 | **0.219** | **0.181/0.543** |
| | contrastive | 12.73 | 13.26 | 12.48 | -3.58 | 0.408 | **0.328/0.747** | 6.92 | 9.34 | 7.82 | **-6.18** | 0.207 | 0.134/0.487 |

Table 1: Results of automatic evaluation on the DailyDialog and LCCC datasets. BAS means the BARTScore (Yuan et al., 2021). BT means the BLEURT score (Sellam et al., 2020).

more local features if we give moderate values of $\delta$ and $\alpha$ (e.g. $\delta$=0, $\alpha$=0.5). To this problem, we empirically scale up the isotropy loss by applying the large value of $\delta$ and the small value of $\alpha$.

# 4 EXPERIMENTS

We evaluate our method on three kinds of dialogue tasks, i.e., multi-turn dialogue response generation (Li et al., 2017), conversational response retrieval (Lowe et al., 2015) and conversational semantic role labeling (Xu et al., 2021). Note that we just focus on open-domain dialogue tasks in this work since previous studies (Jakobovits et al., 2022) revealed that most task-oriented datasets, like MultiWOZ (Budzianowski et al., 2018) are essentially not conversational and contextualized while our method aims to learn these features in dialogue modeling stage.

## 4.1 MULTI-TURN DIALOGUE RESPONSE GENERATION

**Task & Data** Multi-turn dialogue response generation is to automatically produce the human-like response given the dialogue context. Hopefully, we expect the generated responses are diverse, fluent, coherent and informative. We evaluate our approach on DailyDialog (Li et al., 2017) and LCCC (Wang et al., 2020), wherein DailyDialog is a multi-turn open-domain English dialogue dataset and LCCC is a open-domain Chinese short-text conversation dataset.

**Models & Decoding** We compare the vanilla BART and DialoGPT models to their fine-tuned versions with SimCTG and SimDRC. For each model, we conduct the inference using four decoding strategies, including 1) greedy search; 2) beam search (beam size = 5); 3) nucleus sampling ($p = 0.95$) (Holtzman et al., 2019); 4) contrastive decoding ($k = 5$, $\alpha = 0.6$) (Su et al., 2022).

**Evaluation** It is a long-standing problem to accurately and comprehensively evaluate the quality of the generated texts (Celikyilmaz et al., 2020). The traditional n-gram overlap and text matching metrics, such as BLEU (Papineni et al., 2002) and ROUGE (Lin, 2004), are not good choices to open-domain dialogue systems which permit significant diversity and allow multiple plausible outputs for a given input. Therefore, we choose to measure the generation quality over following automatic evaluation metrics, including **BERTScore** (Zhang et al., 2019), **BARTScore** (Yuan et al., 2021), **BLEURT** (Sellam et al., 2020), **Distinct2/4** (Li et al., 2016).

We also conduct the human evaluation with the help of proficient English/Chinese speakers recruited from the crowdsourcing platform. We randomly sample 200 dialogue contexts from DailyDialog and LCCC test sets, respectively. For each dialogue context, we generate responses using different models (vanilla BART, BART+SimCTG and BART+SimDRC) with different decoding strategies (greedy, beam search, nucleus sampling and contrastive decoding). Five annotators are employed to independently evaluate these 2,400 samples by giving a score ranging from 1 to 5 over following aspects, including **fluency**, **informativeness**, **coherence**, and **semantic coverage**[1].

---

[1]More details of human evaluation in Appendix A.

| Model | Method | DailyDialog | | | | LCCC | | | |
|-------|--------|---------|----------------|-----------|----------|---------|----------------|-----------|----------|
| | | Fluency | Informativeness | Coherence | Coverage | Fluency | Informativeness | Coherence | Coverage |
| Agreement | - | 0.65 | 0.56 | 0.71 | 0.72 | 0.66 | 0.78 | 0.70 | 0.68 |
| BART | greedy | 4.62 | 3.48 | 3.72 | 2.50 | 4.65 | 2.84 | 3.58 | 2.09 |
| | beam | 4.59 | 3.51 | 3.62 | 2.50 | 4.47 | 3.05 | 3.32 | 2.10 |
| | nucleus | 4.66 | 3.45 | 3.53 | 2.44 | 4.64 | 2.96 | 3.57 | 2.06 |
| | contrastive | 4.63 | 2.60 | 2.44 | 1.27 | 4.61 | 3.00 | 3.18 | 1.79 |
| SimCTG ($\rho$=0.5) | greedy | 4.63 | 3.50 | 3.69 | 2.51 | 4.62 | 2.87 | 3.63 | 2.05 |
| | beam | 4.65 | 2.54 | 2.47 | 1.26 | 4.61 | 2.96 | 3.14 | 1.79 |
| | nucleus | 4.69 | 3.52 | 3.52 | 2.45 | 4.42 | 2.96 | 3.31 | 2.13 |
| | contrastive | 4.70 | 3.57 | 3.70 | 2.66 | 4.68 | 2.94 | 3.66 | 2.13 |
| SimDRC ($\delta$=0.7, $\alpha$=0.3) | greedy | 4.58 | 2.87 | 3.22 | 1.83 | 4.64 | 3.03 | 3.39 | 1.86 |
| | beam | 4.68 | 3.53 | 3.67 | 2.63 | 4.64 | 2.92 | 3.67 | 2.12 |
| | nucleus | **4.76** | 3.66 | **3.90** | **2.71** | **4.68** | **3.08** | **3.70** | **2.23** |
| | contrastive | 4.62 | **3.69** | 3.77 | 2.61 | 4.50 | 2.95 | 3.27 | 1.99 |

Table 2: Results of human evaluation on the DailyDialog and LCCC datasets.

**Results & Discussion** Table 1 lists the automatic evaluation results of the different methods (i.e. BART, BART+SimCTG and BART+SimDRC)[2] with the different decoding strategies. On both datasets, we can see that SimDRC, especially when equipped with nucleus sampling, achieves the best performance across the board (sign test p < 0.01 compared to SimCTG+contrastive), indicating that SimDRC can produce diverse (high distinct scores), semantically consistent (high BERTScores, BLEURT scores and BARTScores) and informative (high distinct and BARTscores) responses. Inconsistent with the findings of SimCTG where their human evaluation showed that both SimCTG and contrastive decoding could improve the quality of the generated texts on the two datasets in English and Chinese, our automatic evaluation results find the difference that SimCTG+contrastive performs comparably with our method on the LCCC dataset, but performs poorly on the DailyDialog dataset. In contrast, SimDRC constantly performs well regardless of the decoding methods and datasets used, suggesting SimDRC is a more robust solution. We also conduct ablation studies regarding the locality and isotropy training objectives. We find that after removing either locality loss or isotropy loss, the quality of generated texts degrades to a certain extent (details in Section 5.2).

Table 2 summaries the results of human evaluation on DailyDialog and LCCC datasets. The first row indicates the inter-rater agreements calculated by Fleiss' kappa coefficient (Fleiss, 1971). In the rest part, we find that the results of human evaluation are highly consistent with automatic evaluation's, showing that SimDRC+nucleus significantly outperforms baselines across the board. The exception is that SimDRC+contrastive obtains better informativeness score than SimDRC+nucleus on DailyDialog. We think this is reasonable since contrastive decoding encourages the model to generate the response with fewer repetitions and more novel terms. It is also in line with the results of automatic evaluation that SimDRC+contrastive achieves better distinct scores on DailyDialog.

## 4.2 CONVERSATIONAL RESPONSE RETRIEVAL

**Task & Data** Conversational response retrieval (Lowe et al., 2015) is to automatically select the appropriate response from the corpus given a prefix context. Unlike response generation which produces the target sequence word by word, response retrieval finds the potential responses by matching the context representations with the candidate utterance representations. We evaluate our approach on three commonly used response retrieval datasets, including Douban (Wu et al., 2017), E_commerce (Zhang et al., 2018) and Ubuntu (Lowe et al., 2015), wherein Douban is a daily chit-chat dataset while E_commerce and Ubuntu are domain-specific.

**Model & Baselines** As our approach is architecture-agnostic and can be applied to most existing models, we choose the current best-performing response retrieval model, i.e. BERT-FP (Han et al., 2021), as our backbone. BERT-FP is a simple BERT-based retrieval model but benefits from the fine-grained post-training that reflects the characteristics of the multi-turn dialogue. We directly apply SimDRC or SimCTG to the hidden states from the last layer of the original BERT-FP model.

**Results & Discussion** Table 3 summarizes the evaluation results on three datasets. We can see that our approach (BERT-FP + SimDRC) achieves comparable or better performance against the

---

[2]We put the results of DialoGPT models in Appendix C. The findings and conclusions on the DialoGPT models are similar to the BART models'.

| Models | Ubuntu | | | Douban | | | | | | E-commerce | | |
|---|---|---|---|---|---|---|---|---|---|---|---|---|
| | $R_{10}@1$ | $R_{10}@2$ | $R_{10}@5$ | MAP | MRR | P@1 | $R_{10}@1$ | $R_{10}@2$ | $R_{10}@5$ | $R_{10}@1$ | $R_{10}@2$ | $R_{10}@5$ |
| Prev. SoTA* | **0.914** | **0.964** | 0.995 | 0.633 | 0.669 | 0.489 | 0.311 | 0.529 | 0.867 | 0.865 | 0.948 | **0.993** |
| SimCTG($\rho$=0.5) | 0.913 | 0.963 | 0.994 | 0.640 | 0.679 | 0.507 | 0.324 | 0.531 | 0.862 | 0.862 | 0.958 | **0.993** |
| SimDRC($\delta$=0.8 ,$\alpha$=0.6) | 0.913 | 0.963 | **0.995** | **0.643** | **0.681** | **0.510** | **0.326** | **0.536** | **0.872** | **0.869** | **0.959** | 0.991 |
| only w/ locality | 0.913 | 0.963 | 0.994 | 0.637 | 0.680 | 0.514 | 0.325 | 0.532 | 0.856 | 0.868 | 0.959 | 0.992 |
| only w/ isotropy | 0.912 | 0.963 | 0.993 | 0.638 | 0.680 | 0.510 | 0.325 | 0.530 | 0.867 | 0.866 | 0.958 | 0.991 |

Table 3: Evaluation results on the Ubuntu, Douban and E-commerce corpus. * indicates our reproducing results based on the public code released by (Han et al., 2021).

| Method | DuConv | | | NewsDialog | | | PersonalDialog | | |
|---|---|---|---|---|---|---|---|---|---|
| | $F1_{all}$ | $F1_{cross}$ | $F1_{intra}$ | $F1_{all}$ | $F1_{cross}$ | $F1_{intra}$ | $F1_{all}$ | $F1_{cross}$ | $F1_{intra}$ |
| Prev. SoTA | 89.47 | 84.57 | 90.15 | 80.86 | 55.54 | 84.24 | 71.82 | 36.89 | 75.46 |
| SimCTG ($\rho = 0.2$) | 89.28 | 83.17 | 90.27 | 80.86 | 57.28 | 84.18 | 71.50 | 43.45 | 73.83 |
| SimDRC($\delta = 0.5$, $\alpha = 0.2$) | **89.83** | **84.75** | **90.51** | **81.22** | **57.77** | **84.50** | **74.03** | **48.64** | **76.84** |
| only w/ locality | 89.61 | 84.33 | 90.28 | 81.63 | 57.55 | 84.82 | 72.74 | 43.66 | 75.99 |
| only w/ isotropy | 89.30 | 84.95 | 90.09 | 80.35 | 58.18 | 83.57 | 71.43 | 42.06 | 74.91 |

Table 4: Evaluation results on the DuConv, PersonalDialog and NewsDialog datasets.

baselines over all metrics, indicating that SimDRC is also an effective solution to retrieval-based tasks. Specifically, our approach reaches the new state-of-the-art performance on the Douban corpus (sign test p < 0.01), but we find a few performance drops on the Ubuntu and E-commerce corpus. We think this is because there is an extremely small room for performance improvements since the previous best-performing model has achieved over 99% accuracy on $R_{10}@5$. For ablation studies, we find separately employing locality or isotropy also outperforms the baselines.

## 4.3 CONVERSATIONAL SEMANTIC ROLE LABELING

**Task & Data** Conversational semantic role labeling (CSRL) (Xu et al., 2021) is a challenging dialogue understanding task, which extends the standard SRL task (Palmer et al., 2010) to dialogue scenario and aims to find the potential arguments over the entire conversation with the given predicate. Following previous work (Xu et al., 2021), we conduct experiments on three CSRL datasets, namely DuConv, NewsDialog and PersonalDialog. DuConv annotations are split into 80%/10%/10% as train/dev/test sets, while NewsDialog and PersonalDialog are used as out-of-domain test sets.

**Model & Baselines** We adopt CSAGN (Wu et al., 2021) model as our baseline method. CSAGN firstly obtains the dialogue representations from pre-trained BERT, and then employs a predicate-aware module and a conversation structure-aware graph network to capture high-level features. We apply SimDRC or SimCTG on the generated representations from the last layer of BERT.

**Results & Discussion** Table 4 shows the evaluation results on three datasets. We can see that our approach (CSAGN+SimDRC) significantly outperforms all baseline models across the board (sign test with p-value < 0.01), suggesting that SimDRC helps the model to capture both local and conversational features. SimCTG helps the model find more cross-arguments on two out-of-domain test sets while also missing more intra-arguments. This is in line with the characteristics of SimCTG because pushing away all tokens makes the representations more isotropic but less concentrated on local information. Superior to SimCTG, SimDRC improves both locality and isotropy. For ablation studies, we can find 1) individually adopting locality loss also outperforms the baselines; 2) locality consistently enhances the capacity of local feature modeling while isotropy tends to enhance the capacity of cross-turn feature modeling.

## 5 IN-DEPTH ANALYSES

### 5.1 CONVERSATIONAL COHERENCE

A main concern of our approach is that SimDRC may weaken the intrinsic correlations among utterances since it tries to push away the utterance representations and make them discrimina-

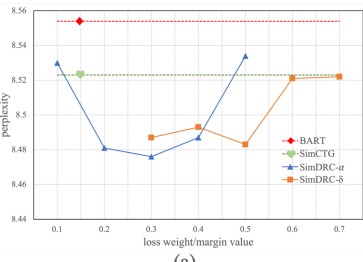 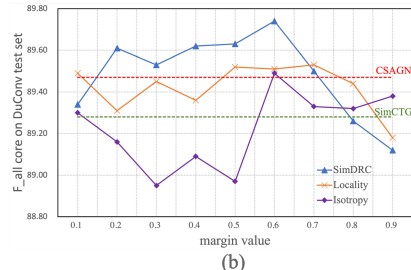

(a)     (b)

Figure 2: (a) The effects of $\alpha$ and $\delta$. SimDRC-$\alpha$/$\delta$ is the average score of all models trained with the candidate $\delta$/$\alpha$. (b) The effect of margin value $\delta$ when $\alpha = 0.5$ on the DuConv test set.

tive. Intuitively, the most straightforward idea of improving the quality of dialogue modeling should be strengthening the connections among utterances, thus making the dialogue more coherent. To this end, we study the conversational coherence in SimDRC-based models. As there are no available methods to automatically measure the conversational coherence based on the contextual representations, we take the average score of the similarities between the dialogue embedding and the utterance embeddings as the coherence score, which is formulated as: **coherence** = $\frac{1}{N}\sum_{i=1}^{N}\frac{\mathbf{v}_{u_i}^T\mathbf{v}_{context}}{\|\mathbf{v}_{u_i}\|\|\mathbf{v}_{context}\|}$, where $\mathbf{v}_{u_i}$ is the utterance representation of utterance $u_i$ and $\mathbf{v}_{context}$ is the dialogue representation. The motivation behind is that the utterances of a coherent dialogue should consistently follow a main topic, which is expressed as the dialogue embedding in our settings.

We calculate the coherence score on the DailyDialog dataset. As shown in Table 5, SimDRC obtains the highest coherence score over other methods while the score of SimCTG is even lower than the vanilla BART model. We think this is because SimCTG constantly focuses on

| Model | BART | SimCTG | SimDRC |
|---|---|---|---|
| Coherence | 0.407 | 0.405 | 0.413 |

Table 5: Conversational coherence scores on DailyDialog.

pushing away all token representations instead of modeling the conversational structure-aware representations, like SimDRC. Moreover, on the CSRL task, SimDRC greatly improves CSAGN's capacity of identifying cross-arguments, consistently showing its ability to enhance conversational coherence.

## 5.2 Effects of Loss weight and Margin value

Here, we explore how the loss weight $\alpha$ and margin value $\delta$ affects the performance of the response generation task. On the DailyDialog dataset, we fine-tune the BART+SimDRC models with different loss weights $\alpha \in [0.1, 0.9]$ and margin values $\delta \in [0.1, 0.9]$ and measure the perplexity on the test set. We experimentally find that the large value of $\alpha$ (e.g. $0.6 \sim 0.9$) and the marginal values of $\delta$ (e.g. $0.1 \sim 0.2, 0.8 \sim 0.9$) would hurt the performance, which is consistent with the discussion in the last paragraph of Section 3.3. To this end, we narrow the candidate sets of $\alpha$ and $\delta$ to $\alpha \in [0.1, 0.5]$ and $\delta \in [0.3, 0.7]$. Figure 2(a) plots the results of the models trained with different hyper-parameters. We can see that SimDRC consistently achieves comparable or better performance than the baselines. And it obtains the best performance when $\alpha = 0.3$ and $\delta = 0.5$.

For the CSRL task, we find the variations of $\alpha$ do not lead to a big weave of the performance. Based on this observation, we fix the value of $\alpha = 0.5$ to study the effects of margin value $\delta \in [0.1, 0.9]$. Figure 2(b) illustrates the results of the models trained with different margin values. We see that SimDRC obtains better scores at the most time. Still, the marginal values of $\delta$ would affect the performance. Furthermore, individually introducing isotropy loss to CSAGN is not an effective solution while only using locality loss performs comparably. The optimal values of $\alpha$ and $\delta$ generally differ from task to task. It would take much effort to search the best hyper-parameters for different tasks. With the extensive experiments in this work, we find that the settings of $\alpha \in [0.2, 0.4]$ and $\delta \in [0.4, 0.6]$ work well for most tasks.

## 5.3 Measurements of Locality and Isotropy Distance

To ensure that SimDRC exactly optimizes the model regarding the properties of locality and isotropy, we measure the locality distance and isotropy distance along with the training process. Specifically,

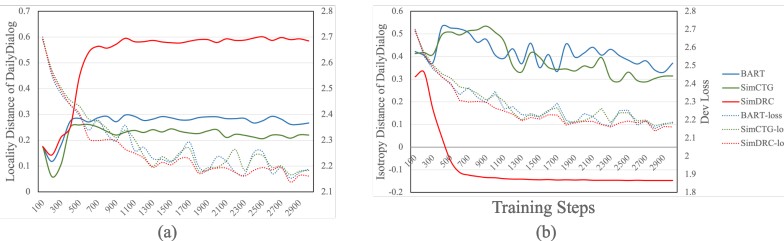

Figure 3: The values of locality distance (a), isotropy distance (b) and development loss calculated by checkpoints in different training steps.

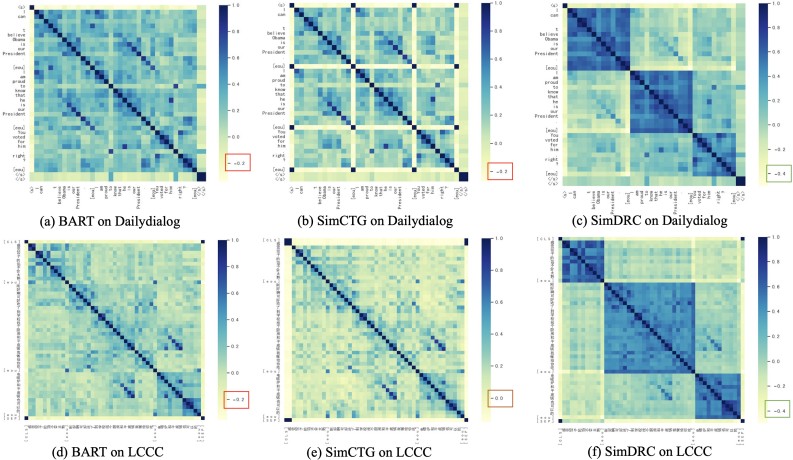

Figure 4: Token similarity matrices of different models trained on DailyDialog and LCCC.

we fine-tune the BART-based models on the DailyDialog training set and measure the locality and isotropy distances on the development set, following the Equation 3 and 5. Figure 3 plots the values of locality, isotropy distances and the development loss at different training stages. We see that SimDRC (red line) obtains much better locality and isotropy distances than the vanilla BART (blue line) and SimCTG (green line), suggesting that SimDRC is essentially working on these two properties. Additionally, larger fluctuations on these two distance values are spotted when training with BART and SimCTG, while the values are steady with SimDRC, especially in terms of isotropy distance. We think this is also the reason that SimDRC has a more smooth development loss curve than baselines.

## 5.4 VISUALIZATION OF TOKEN SIMILARITY MATRIX

In this part, we visualize the token similarity matrices of different models, i.e. vanilla BART, BART+SimCTG and BART+SimDRC. We first fine-tune the models on DailyDialog and LCCC datasets, and then employ the well-trained models to encode the dialogue context and generate the token similarity matrices. Figure 4 showcases the resulted heatmaps on these two datasets. We can see that 1) The token similarities of vanilla BART are relatively high and uniform, i.e. the token is close to most of the other tokens on representations; 2) the token similarities are overall decreased after introducing SimCTG, indicating it alleviates the problem of anisotropy, especially on the LCCC corpus. However, the utterances are still indistinguishable on representations, suggesting that SimCTG also ignores modeling the conversational features; 3) the models with SimDRC generates isotropic and conversational representations, making all utterance discriminative on representations. The observations are in agreement with our motivations and discussions above.

## 6 CONCLUSION AND FUTURE WORK

In this work, we present a simple dialogue representation calibration method to encourage the model to learn isotropic and conversational features in dialogue modeling stage. Experimental results show that our method achieves impressive performance on three dialogue tasks with eight benchmarks. For future work, applying SimDRC into dialogue pre-training would be a promising research direction.

## ACKNOWLEDGEMENTS

We sincerely appreciate the valuable and constructive comments from the reviewers. This work was supported in part by the Changsha Science and Technology Program International and Regional Science and Technology Cooperation Project under Grants kh2201026, the Hong Kong RGC grant ECS 21212419, the Technological Breakthrough Project of Science, Technology and Innovation Commission of Shenzhen Municipality under Grants JSGG20201102162000001, InnoHK initiative, the Government of the HKSAR, Laboratory for AI-Powered Financial Technologies, and the Tencent AI Lab Rhino-Bird Gift Fund with Hong Kong RGC RMGS.

## ETHICS STATEMENT

In this work, we use the publicly released datasets to train/valid/test our models. Generally, these previous works have considered the ethical issues when creating the datasets. For the datasets we used in this work, we have manually checked some samples, and do not find any obvious ethical concerns, such as violent or offensive content. We will also release the source code and the well-trained models along with friendly instructions to support its correct use. However, we still need to emphasize that text generation is not as controllable as we think. It still would generate some novel or unexpected words occasionally. We may take the actions to decrease the generation diversity to alleviate this problem.

## REPRODUCIBILITY STATEMENT

In this work, we present a simple method for dialogue representation calibration. We have provided detailed descriptions of our proposed algorithm in Section 3. For different downstream evaluated tasks, we elaborate the experimental settings and hyper-parameters in Section 4 and Appendix B. All datasets used in this work have been publicly released.

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

## A    HUMAN EVALUATION INSTRUCTIONS

Please rate the quality of the generated response based on the given dialogue context and the target response over following aspects: (1) Fluency; (2) Informativeness; (3) Coherence; (4)Semantic Coverage. We provide some instructions for your rating.

### A.1    FLUENCY

This measures whether the generated text has no formatting problems, capitalization errors or obviously ungrammatical sentences (e.g., fragments, missing components) that make the text difficult to read. The definitions of different scores are:

- **5**: The text is fluent, grammatically correct, and has no errors. It is easy to read.

- **4**: The text is grammatically correct, but has a few spelling or capitalization errors, which does not affect your understanding.

- **3**: The text has minor errors in both grammar and spelling. The errors slightly affect your understanding.

- **2**: The text has major errors in both grammar and spelling. The errors make the text hard to read.

- **1**: The text does not make sense and it is unreadable.

### A.2    INFORMATIVENESS

This measures whether the generated text has diverse, informative, novel or logically related contents. The definitions of different scores are:

- **5**: The text contains very diverse, informative and novel contents. It is enjoyable to read the text.

- **4**: The text contains many informative and novel contents. (Choose this score when you hesitate between 3 and 5.)

- **3**: The text contains some new information but also contains a few repetitions of the context.

- **2**: The text only contains a few informative and new terms. (Choose this score when you hesitate between 1 and 3.)

- **1**: The text is dull, repetitive and has no new information. All contents are from the dialogue context.

### A.3    COHERENCE

This measures whether the generated text is semantically and factually consistent with the dialogue context. The definitions of different scores are:

- **5**: The text is semantically, factually and topically consistent with the dialogue context. All contents of the text are related to the source text or can be inferred from the source.

- **4**: The text is very related to the context but has minor inconsistency or contradictions that do not affect its overall relevance.

- **3**: The text is related to the context but has some obvious inconsistency and contradictions.

- **2**: The text is slightly consistent with the context. Many inconsistency and contradictions to the context can be found.

- **1**: The text is totally inconsistent with the context. It is semantically or factually contradicted to the context.

## A.4 SEMANTIC COVERAGE

This measures how many semantic content units from the target response are covered by the generated text. The definitions of different scores are:

- **5**: All semantic content units of the target text can be found from the generated text. They are semantically consistent.

- **4**: Most of the content units of the target text can be found from the generated text while a few missing units do not affect the overall coverage.

- **3**: Some semantic content units can be found from the generated text but also miss some important units.

- **2**: Most of semantic content units are not covered. Only a few insignificant units can be found in the generated text.

- **1**: The text does not have any overlapping semantic content units with the target text.

We recruit five human workers to annotate 24,000 samples. To make sure the workers are fairly paid, we pay 0.1 dollars for each sample. Therefore, the total amount spent on participant compensation is 2,400 dollars. The annotators take 48 hours to finish the task, suggesting the hourly wage for each worker is 10 dollars.

## B  MORE DETAILS OF THE TASKS

### B.1  MULTI-TURN DIALOGUE RESPONSE GENERATION

**Training**   We fine-tune the models on the DailyDialog and LCCC datasets for 8k and 40k steps, respectively. We use a batch size of 128 and truncate the training samples to a maximum length of 256. The parameters of the models are initialized from Huggingface Libraries (Wolf et al., 2019a) and updated by Adam optimizer (Kingma & Ba, 2015) with a learning rate of 3e-5. The margin values of SimCTG and SimDRC are set to 0.5 and 0.7, respectively. The loss weight $\alpha$ is 0.3. All hyper-parameters are selected from the development set. The training process on the DailyDialog and LCCC datasets takes 0.7 hours and 4 hours on four A100 GPUs, respectively.

**Evaluation**   We conduct automatic evaluations for the response generation task on following metrics: 1) **BERTScore** (Zhang et al., 2019) which calculates the similarities of token representations between the generated response and the target response using the pre-trained BERT model; 2) **BARTScore** (Yuan et al., 2021) which estimates the difficulties of converting the generated text to the reference output by the text generation method; 3) **BLEURT** (Sellam et al., 2020) which is a reference-based text generation metrics that is robust to both domain and quality drifts; and 4) **Distinct2/4** (Li et al., 2016) which computes the generation repetition at different n-gram levels.

### B.2  CONVERSATIONAL RESPONSE RETRIEVAL

**Training**   For each dataset, we fine-tune the model on the training set for 3 epochs, and save the best model according to the performance on the development set. The model parameters are initialized from the post-training checkpoint released by Han et al. (2021) and updated by Adam optimizer with a learning rate of $1e^{-5}$. The margin values in SimCTG and SimDRC are set to 0.5 and 0.8, respectively. The value of $\alpha$ in SimDRC is set to 0.6. All hyper-parameters are selected on the development set. The training process on each dataset takes around 10 hours on a single A100 GPU.

**Evaluation**   Following previous work (Zhang et al., 2018; Han et al., 2021), we use *Recall* (i.e. $R_{10}@k$, $k = (1, 2, 5)$) as our evaluation metric, which indicates the probabilities of whether the correct answer stands in the top $k$ candidates given 10 samples. For the Douban benchmark, we also compute the values of mean average precision (*MAP*) and mean reciprocal rank (*MRR*), and precision at one (*P@1*) since the context in this dataset may contain multiple positive responses.

### B.3 CONVERSATIONAL SEMANTIC ROLE LABELING

**Training**    We follow the previous work (Wu et al., 2021) and solve the CSRL task as the sequence labeling problem. We keep the training settings same to CSAGN's. The parameters of the model are initialized from the pre-trained BERT, and updated by Adam optimizer with a linear learning rate schedule. The margin values of SimCTG and SimDRC are set to 0.2 and 0.5, respectively. The loss weight $\alpha$ in SimDRC is set to 0.2. All hyper-parameters are selected on the development set. The training process on the DuConv training set takes around 2 hours on two A100 GPUs.

**Evaluation**    Following (Xu et al., 2021; Wu et al., 2021), we report the $F1_{all}$, $F1_{intra}$ and $F1_{inter}$ scores over the (predicate, argument, label) tuples. The arguments are categorized into two types, i.e., intra-arguments and cross-arguments, according to whether the argument appears in the same turn with the predicate or not. Therefore, we calculate the $F1_{intra}$ and $F1_{inter}$ scores on intra- and cross-arguments, respectively.

### B.4 RESULTS OF DIALOGPT ON RESPONSE GENERATION TASK

## C MORE EVALUATION RESULTS

| Model | Method | BERTScore | | | BARTScore↑ | BLEURT↑ | Dis2/4↑ |
|---|---|---|---|---|---|---|---|
| | | P↑ | R↑ | F↑ | | | |
| DialoGPT | greedy | 12.13 | 10.22 | 10.96 | -3.82 | 0.382 | 0.303/0.695 |
| | beam | 12.18 | 12.63 | 11.71 | -3.90 | 0.383 | 0.300/0.671 |
| | nucleus | 12.22 | 12.65 | 12.05 | -3.70 | 0.386 | 0.306/0.692 |
| | contrastive | 10.14 | 11.92 | 10.56 | -4.19 | 0.247 | 0.288/0.653 |
| SimCTG ($\rho$=0.6) | greedy | 11.31 | 9.69 | 10.00 | -3.98 | 0.371 | 0.271/0.622 |
| | beam | 12.01 | 12.46 | 12.15 | -3.73 | 0.375 | 0.273/0.632 |
| | nucleus | 10.54 | 11.63 | 10.65 | -3.79 | 0.366 | 0.269/0.627 |
| | contrastive | 12.12 | 12.62 | 12.22 | -3.71 | 0.375 | 0.274/0.631 |
| SimDRC ($\delta$=0.5, $\alpha$=0.2) | greedy | 12.05 | 12.10 | 12.01 | -3.90 | 0.322 | 0.271/0.639 |
| | beam | 12.63 | 13.75 | 12.65 | -3.69 | 0.385 | 0.277/0.634 |
| | nucleus | **12.74** | **13.88** | **12.78** | -3.65 | **0.399** | 0.317/0.713 |
| | contrastive | 12.62 | 13.15 | 12.45 | **-3.64** | 0.392 | **0.322/0.744** |

Table 6: Results of automatic evaluation on the DailyDialog dataset.

Table 6 shows the results of DialoGPT models on DailyDialog. Since there are no suitable DialoGPT models for Chinese, we only evaluate on the DailyDialog dataset here.

## D MORE IN-DEPTH ANALYSES

### D.1 VISUALIZATION OF SELF-ATTENTION WEIGHTS

In this part, we visualize the self-attention weights of vanilla BART and BART+SimDRC trained on DailyDialog. As shown in Figure 5, we can see that 1) in vanilla BART, all tokens are primarily attended to the dialogue representative token, i.e.  in our cases. This would lead to the problem that all tokens receive much the same information and be nearby to each other on representations. This is exactly the problem of *anisotropy*; 2) with the aid of SimDRC, the tokens are encouraged to be concentrated on the tokens within the same utterance and discriminative to tokens in different utterances. Therefore, we believe that SimDRC also essentially learns locality and isotropy on attention weights, while it explicitly calibrates on token embeddings.

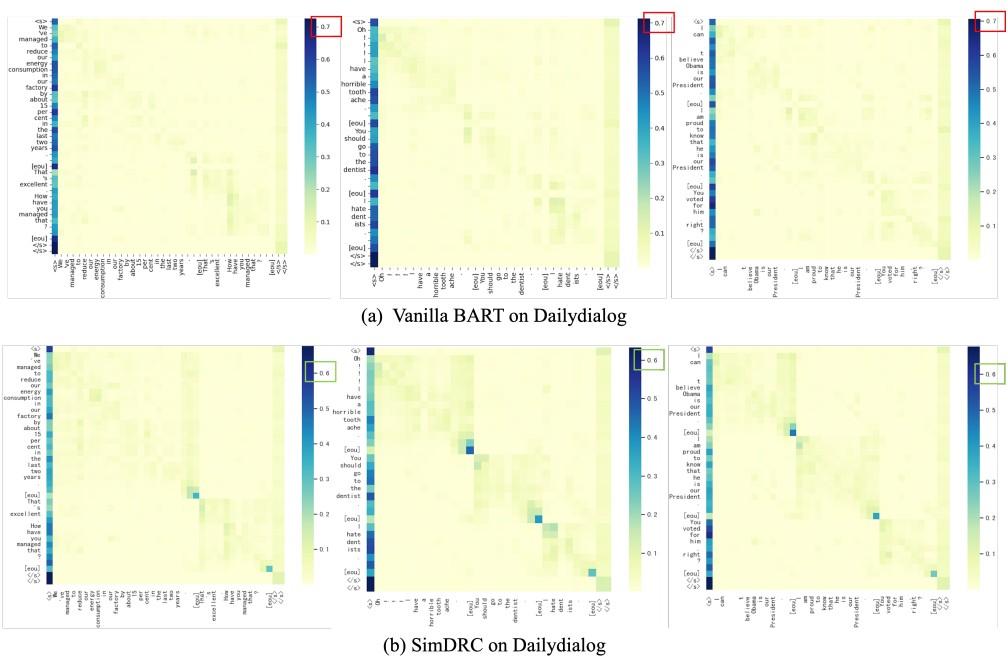

Figure 5: Attention weights visualization of different models trained on DailyDialog.

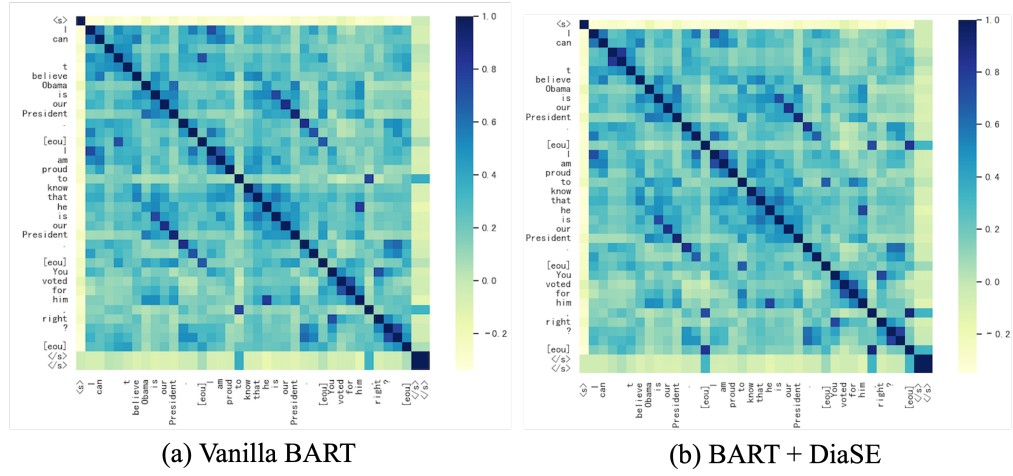

Figure 6: Comparison of vanilla BART and BART with dialogue state embeddings.

## D.2 DIALOGUE STATE EMBEDDING VS. SIMDRC

A straightforward idea to distinguish different utterance might be simply adding a dialogue state embedding (DiaSE) (Wolf et al., 2019b; Xu et al., 2021) to token embeddings. To this end, we introduce an additional randomly initialized dialogue state embedding layer into vanilla BART to differentiate different turns. During encoding, these additional embeddings are added with word embeddings and positional embeddings. As shown in Figure 6(b), even with the help of the turn indicator, the problem of anisotropy still remains. We think the reason is that DiaSE is essentially another kind of "segment embedding" which is not capable of capturing deep semantic or conversational features.

## D.3 EXAMPLES OF GENERATED TEXTS

| Context | SimDRC | Label |
|---|---|---|
| **A**: 刺猬很可爱！以前别人送了只没养，味儿太大！
**B**: 是很可爱但是非常臭。 | 是的，我家的刺猬也是，太臭没办法养 | 是啊，没办法养 |
| **A**: 还发说说感冒早睡的人都几点了
**B**: 哈哈，睡不着
**A**: 看电视 | 哈哈，看电视也睡不着 | 看你 |
| **A**: 都有什么好玩的哇
**B**: 没啥好玩的、一点儿意思都没有
**A**: 那跟沈阳差不多，还是大连好 | 大连好玩的地方太多了！ | 嗯呐、大连比这地儿好多了 |
| **A**: puma的板鞋就挺好看的
**B**: 我看了，我这里差不多随处就能看到一双 | 哈哈哈，我这里也有一双，但是不好看 | 但是确实很百搭 |
| **A**: 当你遇到一个不讲道理的女票又不忍心吵架时，你会发现沉默挺好的 | 不讲道理的女朋友是不会和你吵架的 | 讲道理，如果说自己女朋友无理取闹，当初为什么又要追她呢？其实并不矛盾。 |

Table 7: More generated examples on LCCC.

| Context | A: Have you got any experience in advertising? |
|---|---|
| | B: Yes, I have been working in the Public Relations Section of a company in the past two years. I plan the advertising campaign and cooperate the work of artists and typographers. |
| | A: Do you have any practical experience as a secretary? |
| | B: Yes, sir. I acted as a secretary in an insurance company. |
| | A: Have you had any experience with computer? |
| BART | Yes, I have used a computer several times in the past two years. I've learned how to use it very well. |
| SimCTG | Yes, I have worked in a computer company for two years and I have a computer of my own. |
| SimDRC | Yes, I have worked in a computer company for two years and I have used Microsoft Word and Excel. |
| Context | A: Listen, Karen, I need your help. I don't know anyone here yet. |
| | B: I'm glad to help you. What's wrong? |
| | A: My mother-in-law just went into the hospital in L.A. Hank and I will be flying there tonight. |
| | B: I'm sorry to hear it. What's wrong with her? |
| | A: Doctors aren't sure yet. But the real problem is Suzy. She has a bad cold, and we don't want to take her with us. But we don't know who can stay with her. |
| | B: You need a babysitter. |
| | A: Yes, I'd ask Jill, the girl I've had before, but I need someone overnight.Maybe even for two nights. |
| BART | That's a good idea. I'll ask Jill. She's very good. |
| SimCTG | I'm sorry to hear that, but I don't think Jill would be able to do that. |
| SimDRC | I don't think Jill would be able to do it. She's not good at babysitting because she's only 17. |
| Context | A: 911 emergency. What is the problem? |
| | B: I would like to report a break-in. |
| | A: Where was the break-in? |
| | B: It happened at my house. |
| | A: Was anything stolen? |
| | B: I haven't checked yet. |
| | A: Why do you think there was a break-in? |
| BART | I don't know, but I think it was a robbery. |
| SimCTG | There was a break-in at my house last night. |
| SimDRC | I believe it was a break-in because the door was left open. |
| Context | A: Excuse me. I have an appointment with Mr. Li at nine. May I come in? |
| | B: Yes, come in please. I am Mr. Li. You must be My Liu, right? |
| | A: Yes, I am My Liu. Thanks. |
| | B: I'd like to start this interview with some questions. Why do you think you are qualified for this position? |
| BART | I am very interested in this position and I am very competent in this area. |
| SimCTG | I think I am qualified for this position because I have worked in this company for more than five years and I have experience in this field. |
| SimDRC | Well, first of all, I think I am a good team-player and I have a lot of experience in this line of work. Second, I enjoy working with people and have a good working relationship with my colleagues. Finally, I am very interested in this position. |
| Context | A: How do you get road news? |
| | B: I've a driver mate in my car! |
| | A: So you have all the news on broadcast? |
| BART | Yes, I've got all the news in my car! |
| SimCTG | Yes, I've got it on the TV, too. |
| SimDRC | Yes, I've got all the news on the radio in my car. |

Table 8: More generated examples on DailyDialog.

