# OpenReview forum: "Learning Locality and Isotropy in Dialogue Modeling"
_ICLR.cc/2023/Conference — ICLR 2023 poster_

### Official Review · Reviewer_uyyF · 2022-10-23

**Confidence:** 3
**Correctness:** 3
**Technical Novelty And Significance:** 2
**Empirical Novelty And Significance:** 3
**Recommendation:** 6

**Clarity, Quality, Novelty And Reproducibility:**

Clarity, Quality, Novelty: good.

Reproducibility: lack implementation details (e.g., model training), code is not available.

**Strength And Weaknesses:**

Strengths:

- The paper is well organized and easy to follow.
- The authors presented extensive experimental results on 3 different dialogue tasks with 2 different language model (BART and DialoGPT).

Weakness:

- I am not fully convinced that "most task-oriented datasets, like MultiWOZ are essentially not conversational and contextualized" as stated in the paper. There are plenty of multi-turn task-oriented datasets (e.g., SGD, STAR, RiSAWOZ, BiTOD, CrossWoz ). It would make the paper stronger if authors could show the advantage of the proposed method in some task-oriented datasets.

**Summary Of The Paper:**

This paper introduces SimDRC for dialogue representation learning. SimDRC capture the dialogue structure by locality loss and isotropy loss. The locality loss maximizes the cosine similarity of the representations of the tokens within an utterance, while the isotropy minimizes the cosine similarity of the representations of the tokens from different utterances. The authors applied SimDRC to fine-tune BART on 3 different dialogue tasks, i.e., multi-turn dialogue response generation, conversational response retrieval, and conversational semantic role
labeling. The experimental results showed that SimDRC improved the naive fine-tuning strategy in terms of automatic and human evaluation metrics.

**Summary Of The Review:**

This paper proposed a novel method to improve dialog representation learning. But the experiments are only limited to open domain dialogue datasets.

---

> ### Author Response · Authors · 2022-11-12
> **Response to Reviewer uyyF**
>
> Thanks for your valuable review comments.
>
> ***Q1.*** It would make the paper stronger if authors could show the advantage of the proposed method in some
> task-oriented datasets.
>
> ***A1.*** Yes, we have also tested on task-oriented dialogue datasets when we selected the downstream dialogue
> understanding tasks. We reproduced the current best-performing dialogue state tracking model, i.e., PPTOD[1], to
> finish the DST task on MultiWoz 2.0 and 2.1. The results are as follows:
>
> | **Model** | **Joint ACC on MWZ2.0** | **Joint ACC on MWZ2.1** |
> | :----- | :----: | :----: |
> | PPTOD | 53.36 | 57.05 |
> | PPTOD + SimDRC | **53.38** | **57.09** |
>
> We found SimDRC could consistently improve the performance of DST, but not very significant. So, we did not report these
> results in the paper. Another reason that we exclude the task-oriented dialogue system as our targeted tasks is that the
> existing dialogue models, such as TripPy and CHAN, for task-oriented dialogue, heavily rely on high-level features,
> like slot-turn alignment and slot-value alignment. There are some specific patterns to solve the task-oriented dialogue.
> The fundamental dialogue modeling is not as critical as we expect in these tasks compared to open-ended dialogue tasks.
> We will explore how to make SimDRC work well on task-oriented datasets in our future work.
>
> [1] Multi-Task Pre-Training for Plug-and-Play Task-Oriented Dialogue System
>
> ***Q2.*** Reproducibility: lack implementation details (e.g., model training), code is not available.
>
> ***A2.*** We have provided implementation details in Appendix B. We will also publicly release our code and checkpoints
> upon acceptance.

---

> > ### Comment · Reviewer_uyyF · 2022-12-09
> > **TOD Results**
> >
> > I am glad that authors have conducted experiments on ToD datasets. Based on the updated results, I decided to stick to my original score. Although the improvement is not significant, it would be beneficial to analyze the improved cases in MultiWoz to understand why SimDRC is helpful in these cases.

---

### Official Review · Reviewer_uSDJ · 2022-10-24

**Confidence:** 4
**Correctness:** 2
**Technical Novelty And Significance:** 3
**Empirical Novelty And Significance:** 3
**Recommendation:** 6

**Clarity, Quality, Novelty And Reproducibility:**

The paper is well motivated and well written with appropriate illustrations to ensure clarity. The idea of using locality and isotropic constrained to added conversational anisotropy problem is new to the best of my knowledge. However, the sensitivity to training hyperparameters raises concerns for reproducibility.

**Strength And Weaknesses:**

Pros:
The use of locality and isotropic constraints is adequately demonstrated as show in the embedding similarity heat map in Figure 1&4. When applied to response generation, the technique was able to improve the vanilla BART model on both DailyDialog and LCCC datasets.

Con:
Most of the cons are already mentioned in section 5 under in-depth analysis.
Conversational Coherence - One would expect the isotropy between utterance representations to reduce the conversational coherence. However, the analysis showed that SimDRC is actually more conversationally coherent than the vanilla BART and SimCTG. While this is encouraging, it is not clear how the dialog representation v_{context} is obtained in the article since dialog representation is not explicitly stated in the problem formulation. Can the author(s) explain how this is calculated? Would be great if the coherent score can be evaluated on Ubuntu dialog also.

Sensitivity to loss weight and margin value -  The optimal values of these hyper-parameters seem to be dataset dependent which make it unreliable. However, the authors providing ranges of value for each task alleviates this problem to a considerable extent. Nevertheless, it would still be desirable to have more robustness over these hyper-parameter values.

**Summary Of The Paper:**

The paper proposes techniques for generation dialog representation that is both local and isotropic in order to address the problem of anisotropy in conversation modeling. The paper attempted to apply locality and isotropic constraints at both the token and utterance levels. The locality constraint ensures that each token representations is close to the utterance representation while the isotropic constraint ensures that the utterance representations are pushed away from each other. The proposed representation was then applied to 3 dialog related tasks including multi-turn response generation, response retrieval and dialog semantic role labeling. The paper is well written and well motivated. The paper is also easy to follow.

**Summary Of The Review:**

The main idea of the paper seems novel and reasonable. However, there's concern with some of the conclusions drawn by the paper especially the broad applicability to different dialog modeling related tasks.

It is not clear to me that the isotropic constraint did not impact the conversational coherence despite the results shown in Table 5. Would be great to understand how the dialog representation is calculated and if explicitly adding to the problem formulation to ensure conversational coherence would be more beneficial.

Also, the sensitivity for training hyper-parameter raises concerns for reproducibility. I wonder if using contrastive learning loss would have been better than using max-margin losses, which would minimize the number of hyperparameters to be searched for.

---

> ### Author Response · Authors · 2022-11-12
> **Response to Reviewer uSDJ**
>
> Thanks for your valuable review comments.
>
> ***Q1.*** it is not clear how the dialog representation v_{context} is obtained in the article since dialog
> representation is not explicitly stated in the problem formulation.
>
> ***A1.*** As shown in Equation(1), a special token [CONTEXT] is inserted as the last token of the whole dialogue. We
>  directly take the token representation of [CONTEXT] as the dialogue representation $v_{context}$.
>
> ***Q2.*** Would be great if the coherent score can be evaluated on Ubuntu dialog also.
>
> ***A2.*** We report the coherent score on Ubuntu dialogue here.
>
> | **Model** | **Han et. al, 2021** | **SimCTG** | **SimDRC** |
> | :----- | :----: | :----: | :----: |
> | Coherence | 38.49 | 37.91 | **38.52** |
>
> We can obtain similar results to Dailydialog's. SimDRC achieves the best coherence score compared to the vanilla
> model and SimCTG, while SimCTG has the lowest coherence score since it tries to push away all representations.
> Although SimDRC attempts to enlarge the gaps among different utterances, it constantly follows the main topic of the dialogue.
>
> ***Q3.*** Sensitivity to loss weight and margin value
>
> ***A3.*** Yes, the optimal values of $\alpha$ and $\delta$ generally differ from task to task. It would take much effort
> to search for the best hyper-parameters for different tasks. After our extensive experiments, we find that performance
> improvements can always be obtained if $\alpha$ is set to 0.2 ~ 0.4 and $\delta$ is set to 0.4 ~ 0.6. So, to find the optimal
> values of these two hyper-parameters, there are just 3 $\times$ 3 = 9 pairs of experiments. We think this is acceptable.

---

> > ### Comment · Reviewer_uSDJ · 2022-12-09
> > **The response is not convincing but I keep the score as is.**
> >
> > I have read the author's response and based on my understanding, the coherence score calculation seems like an after thought. It should have been straight forward to include the distance between the [context] the utterance delimiter embeddings in the original formulation.
> >
> > Also, the need for hyper-parameter search would lead to reproducibility issues. However, the ideas presented are novel and the ablation studies have been exhaustive. The authors have also taken time to attempt to address other reviewers' feedback, sometimes with additional experiments. Hence the decision to leave the scores unchanged.

---

### Official Review · Reviewer_bCi9 · 2022-10-25

**Confidence:** 4
**Clarity, Quality, Novelty And Reproducibility:** The paper is generally well written, …
**Correctness:** 2
**Technical Novelty And Significance:** 1
**Empirical Novelty And Significance:** 1
**Recommendation:** 6

**Strength And Weaknesses:**

The motivation of this paper is not convincing. I think the inter-speaker correlation can not be well-modeled by the generation model. It would be better to use the social network to enhance the generation model rather than use the model itself.

It is not novel to define a distance and add it to the model via a loss function.

By using the locality loss, and isotropy loss, the token-level similarity will be revised to the pattern shown in Figure 1 (c). But I do not think the inter-speaker correlations and conversational structure information are well-captured by the generation model. There is a gap between the motivation and the proposed model.

**Summary Of The Paper:**

The paper focuses on the problem that the existing large language models fail to learn the dialogue-specific features. The paper tries to solve the locality and isotropy problem for dialogue generation modes by encouraging the model to aggregate the representation of tokens within an utterance and push away the representation of distinct utterances.

**Summary Of The Review:**

The motivation of this paper is not convincing. There is a gap between the motivation and the proposed model.

---

> ### Author Response · Authors · 2022-11-12
> **Response to Reviewer bCi9**
>
> Thanks for your valuable review comments.
>
> ***Q1.*** The motivation of this paper is not convincing. I think the inter-speaker correlation can not be well-modeled
> by the generation model.
>
> ***A1.*** We use BART, an encoder-decoder framework model, to demonstrate the anisotropy problem of existing dialogue
> models. Although BART is a generative model, we just leverage its encoder part to obtain the token similarity heatmaps.
> Besides the response generation, in the other two experiments, our backbone model is BERT, which is not a generative model.
> Furthermore, we do not think the conversational features cannot be well-modeled by the generative models. A bunch of
> previous studies[1,2,3] have shown that generative models are also good participants in conversations.
>
> [1] Dialogpt: Large-scale generative pre-training for conversational response generation
>
> [2] Plato: Pre-trained dialogue generation model with discrete latent variable
>
> [3] Dialoglm: Pre-trained model for long dialogue understanding and summarization
>
> ***Q2.*** It would be better to use the social network to enhance the generation model rather than use
> the model itself.
>
> ***A2.*** There are various ways to enhance cross-turn relations in conversations. Incorporating the social
> network can be an alternative way, but it is not practical since this approach heavily relies on external knowledge
> to build the social network. Actually, for most academic dialogue datasets, like Dailydialog and LCCC, it is very hard
> to construct the social network on those data.
>
> Conversely, directly adjusting the feature space is a more practical and effective approach. Note that we are not
> intended to extract the explicit conversational features, like speaker relations and utterance dependencies, from the
> model. We expect the model can implicitly learn the conversational features on representations with the aid of our
> proposed method.
>
> ***Q3.*** It is not novel to define a distance and add it to the model via a loss function.
>
> ***A3.*** We summarize our contributions as follows:
>
> 1. We are the first one to reveal the isotropic and conversational problems in existing dialogue modeling methods.
>
> 2. To address the above problems, we identify and formulate two properties in dialogues, i.e., *locality* and *isotropy*.
> We are also the first work that quantitatively measures how much local and isotropic information are preserved in representations.
>
> 3. By directly optimizing on the above two properties, we achieve new state-of-the-art performance on three dialogue
> downstream tasks across eight datasets.

---

> > ### Comment · Reviewer_bCi9 · 2022-12-10
> > **Novelty Problem**
> >
> > After reading all the reviews and rebuttals, I think even though designing a loss function to model the locality and isotropy is not novel, the paper presents a new perspective that improves the informativeness and discrimination of the conversational model without using the additional resource. I am fine with the novelty problem so I increase my score.

---

> ### Author Response · Authors · 2022-11-21
> **Make-up Response to Reviewer bCi9 (Part 1/2)**
>
> Regarding your questions, we would like to provide more clarifications.
>
> ### Motivations
> As written in the paper, we train a response generation model using the standard *dialogue* modeling method on *dialogue* data. But unfortunately, we found, as shown in Figure 1(a), the similarities of distinct tokens are relatively high regardless of whether the tokens are within an utterance or not. We claim this is the problem of *anisotropy* and *missing conversational features*. The problem of anisotropy has been clearly explained in our paper and previous work [1,2], which is that features occupy a narrow cone in the vector space. As for the problem of *missing conversational features*, we defined it as **utterances being indistinguishable on representations**. More specifically, we say utterances are distinguishable on representations when the representations of tokens within an utterance are close to voice a concentrated idea of the utterance, and the representations of different utterances are discriminative and isotropic to convey the maximal information of the dialogue. We give such a definition of *conversational feature* from two aspects: 1) human speaking behaviour, is that humans pay more attention to the central idea of the utterance rather than how the utterance is organized by words when humans utter, and humans also prefer to express more information with fewer utterances[3]; 2) the results of data analyses. We randomly sampled 200 pieces of data from Dailydialog and LCCC, respectively. Then we manually annotated whether an utterance talks about more than one topic and whether two utterances in a dialogue express the same meaning. The results showed that around 75% of utterances focus on *ONE* concentrated topic, and 82% of conversations do not have more than two utterances that express the same meaning. We believe this observation is highly consistent with our proposed method. We will also add this result in our final revision.
>
> Regarding your concern that *the inter-speaker correlation and conversational structure information can not be well-modeled by the generation model.*, we clarify our understanding from the following perspectives:
>
> 1. There are two kinds of features, i.e., inter-speaker correlation and conversational structure information. We are more focused on the **conversational structure information** in this work. As mentioned in Section 1 and Section 2, our goal is to produce more isotropic and conversational contextual representations. Based on our definition of *conversational features* (the bold text in the previous paragraph), we consider the conversations consisting of distinguishable utterances are well-structural conversations. According to Figures 1&4, the experimental results and the analyses (conversational coherence), our method is capable of producing conversational contextual representations. Therefore, we believe that conversational structure information can be well-modeled by our method. For *inter-speaker correlations*, it is not the focus of this work since we do not incorporate any speaker information into the model. Maybe the words in the first paragraph lead to your misunderstanding about our main focus. We will revise the words to state our focus clearly.
>
> 2. Concerns about the "generation model". In other words, it should be the question of *is the generative model capable of capturing dialogue features?*. A set of previous studies[4,5,6] have shown that the generative model trained with generative tasks can be an excellent speaker in conversations. More specifically, BART is essentially an encoder-decoder based model where the encoder is responsible for capturing the dialogue features and the decoder generates the tokens based on the encoder outputs. If the encoder-only model, like BERT, is believed to capture dialogue features, the encoder-decoder model, like BART, should be equally treated since it also has an encoder part. Additionally, we would like to note that we not only use the generative models to model the conversational features (the response generation task), but also experiment with encoder-based models on the retrieval and understanding tasks. The results on all downstream tasks consistently show that our proposed method absolutely builds more isotropic and conversational contextual representations.
>
> [1] A contrastive framework for neural text generation, NeurIPS 2022.
>
> [2]  How contextual are contextualized word representations? comparing the geometry of bert, elmo, and gpt-2 embeddings, EMNLP 2019.
>
> [3]  Spatiotemporal dynamics of orthographic and lexical processing in the ventral visual pathway, Nature Human Behaviour 2021.
>
> [4] Dialogpt: Large-scale generative pre-training for conversational response generation
>
> [5] Plato: Pre-trained dialogue generation model with discrete latent variable
>
> [6] Dialoglm: Pre-trained model for long dialogue understanding and summarization

---

> ### Author Response · Authors · 2022-11-21
> **Make-up Response to Reviewer bCi9 (Part 2/2)**
>
> ### For your suggestion, "it would be better to use the social network to enhance the generation model rather than use the model itself."
>
> Yes, we totally agree that applying social networks can enhance the dialogue model, especially for modeling better
> speaker features in dialogue models. However, it is not practical for most datasets since we cannot obtain the social
> information of speakers. To protect the user privacy and avoid ethical issues, it is very hard to obtain the social information
> of speakers and build the social networks based on the existing datasets, like Dailydialog and LCCC.
>
> In contrast, we use the model itself to calibrate its vector space for better conversational feature learning. It is a
> more practical way to enhance the dialogue model. Although it does not model any explicit dialogue-specific features,
> like speaker relations or utterance dependencies, it also could implicitly capture conversational structural features,
> like making utterance distinguishable on representations. Our method is easy to follow and reused since it does not
> rely on any external information or toolkits.
>
> ### Novelty
>
> We believe our method is novel enough. Other three reviewers also agree with the novelty of our work. We would like to
> clarify the novelty and contributions of this work as follows:
>
> 1. We are the first one to reveal the isotropic and conversational problems in existing dialogue modeling methods based
>    on the representations. We think this is a new view to dialogue community that could help researchers to find more
>    issues of existing dialogue models, or create new dialogue evaluation metrics based on the representations rather
>    than the explicit dialogue-specific features.
>
> 2. We identify and formulate two new properties in dialogues, i.e., *locality* and *isotropy*. We are also the first
>    work that quantitatively measures how much local and isotropic information are preserved in representations. We consider
>    these two properties could be further used to dialogue evaluation.
>
> 3. By directly optimizing on the above two properties, we achieve new state-of-the-art performance on three dialogue
>    downstream tasks across eight datasets.
>
> Thanks for your review comments again. Sincerely wish that our response can well settle down your concerns. If you have any other questions or concerns, please let us know. We are pleased to discuss them with you.

---

### Official Review · Reviewer_YMbd · 2022-10-25

**Confidence:** 5
**Correctness:** 4
**Technical Novelty And Significance:** 4
**Empirical Novelty And Significance:** 3
**Recommendation:** 8

**Clarity, Quality, Novelty And Reproducibility:**

As I said in the strength of the paper, it is well written and it articulated very well the new ideas and the new methods, experiments and results, as well as the analysis. It is novel and it will benefit the ICLR audience.
It is also a good extension of the state-of-the-arts.

**Strength And Weaknesses:**

Strength:

This paper defines a new approach for generated dialogue representation calibration based on locality and isotropy. The experiments showed a consistent performance gain in 3 conversational tasks.
The literature study in the paper was complete and the analysis of the experiments was also in-depth.
The paper was organized very well with good articulation and the analysis of the technical problems, the methodology and the corresponding results.

Weakness:

SimCTG seems to be the major focus that the paper is comparing SimDRC with. The paper articulated the difference between the two methods in section 2 (token-level vs. multi-granularity levels), and the results of the experiments in section 4, as well as the conversation coherence analysis in section 5.1. It is better if the authors could include the following analysis:

a) why is SimDRC a better methods that SimCTG in dialogue related tasks? what is missing in SimCTG that make it is not suitable for conversational tasks?

b) what other tasks will SimDRC be good for other than the 3 tasks included in the paper? Is it good for document generation tasks?

**Summary Of The Paper:**

The paper focuses on the problem of how to make it more conversational for the generated dialogue representation from Transformers or large-scale pretrained language models. The authors studied related literatures on the issues that dialogue modeling is not isotropic and conversational, and analyzed the recent representation calibration approaches. Based on these, the authors proposed a new dialogue representation calibration method, namely SimDRC focusing on locality and isotropy to justify the challenges.

The paper first gave the definitions of locality and isotropy in dialogue modeling, then defined the locality loss, isotropy loss and SimDRC loss to adjust feature space to model the dialogue representation.

Then the paper applied the new approach on the state-of-the-art models for 3 dialogue tasks: multi-turn dialogue response generation, conversational response retrieval, conversational semantic role labeling. The results of the experiments demonstrated that by combining SimDRC and state-of-the-art models in these tasks, better results have been achieved across all the 3 tasks than other related methods.
The analysis of conversational coherence, effects of loss weight and margin value, the measurements of locality and isotropy as well as the visualization of token similarity metrics were presented.

**Summary Of The Review:**

This paper created a new approach for generated dialogue representation calibration based on locality and isotropy, demonstrated a consistent performance gain in 3 conversational tasks. It was organized very well to articulate the new methods, experiments and results. It is a good extension of the state-of-the-arts.

---

> ### Author Response · Authors · 2022-11-12
> **Response to Reviewer YMbd**
>
> We sincerely appreciate your valuable review comments.
>
> ***Q1.*** why is SimDRC a better methods that SimCTG in dialogue related tasks? what is missing in SimCTG that make it
> is not suitable for conversational tasks?
>
> ***A1.*** SimCTG is a **token-level** contrastive framework which tries to push away the representations of **all**
> distinct tokens, regardless of whether the tokens are within a sentence. This is radical since the words within a
> sentence tend to express similar meanings at the most time, especially in conversations.
>
> For example, there is a conversation,
>
> *A: I planned to withdraw money from the bank(1). Unfortunately, when I arrived, I found the bank(2) was closed today.*
>
> *B: Sounds terrible. You can come play with me. I am walking along the river bank(3) now.*
>
> When a word appears in different sentences, like bank(1) and bank(3) in the above example, they might have different
> meanings. Conversely, at the most time, the same word within a sentence should be more likely to express the same
> meaning, like bank(1) and bank(2).
>
> ***Q2.*** what other tasks will SimDRC be good for other than the 3 tasks included in the paper?
>
> ***A2.*** Yes, we have also tested on task-oriented dialogue datasets when we selected the downstream dialogue
> understanding tasks. We reproduced the current best-performing dialogue state tracking model, i.e., PPTOD[1], to
> finish the DST task on MultiWoz 2.0 and 2.1. The results are as follows:
>
> | **Model** | **Joint ACC on MWZ2.0** | **Joint ACC on MWZ2.1** |
> | :----- | :----: | :----: |
> | PPTOD | 53.36 | 57.05 |
> | PPTOD + SimDRC | **53.38** | **57.09** |
>
> We found SimDRC could consistently improve the performance of DST, but not very significant. So, we did not report these
> results in the paper. Another reason that we exclude the task-oriented dialogue system as our targeted tasks is that the
> existing dialogue models, such as TripPy and CHAN, for task-oriented dialogue heavily rely on high-level features,
> like slot-turn alignment, and slot-value alignment. There are some specific patterns to solve the task-oriented dialogue.
> The fundamental dialogue modeling is not as critical as we expect in these tasks compared to open-ended dialogue tasks.
>
> ***Q3.*** Is it good for document generation tasks?
>
> ***A3.*** As we focus on the dialogue modeling in this work, we did not try document generation tasks. We are verifying
> the effectiveness of SimDRC on these tasks, but it might take some time to obtain the results.
>
> [1] Multi-Task Pre-Training for Plug-and-Play Task-Oriented Dialogue System

---

### Decision · Program_Chairs · 2023-01-20

**Decision:**

Accept: poster

**Justification For Why Not Higher Score:**

The scope of this work is relatively narrow, as the paper is specific to dialog modeling and might have a reduced audience. Note that the method improved upon (SimCTG) is more generally applicable to text generation.

**Justification For Why Not Lower Score:**

While this is a targeted paper on dialog modeling, the analyzes are relatively in depth. The main concerns of the reviewers have been addressed.

The idea of using both locality loss and isotropy loss (apparently new) could have application in other work on representation learning.

**Metareview: Summary, Strengths And Weaknesses:**

The paper is concerned with the problem of anisotropy noted in prior work on Transformer models, which causes the cosine similarity of randomly-picked token pairs to be relatively high. While prior works (e.g., SimCTG) help remedy anisotropy, they are generally insensitive to conversational structure, e.g., causing representations of identical tokens in different utterances to be similar. To improve upon SimCTG, the authors contribute a simple yet effective dialogue representation calibration method that encourages the model to learn isotropic and conversational features. Experiments on three open-domain downstream tasks show significant improvements over SimCTG and baselines (e.g., BART). While the paper’s contributions are specific to dialog, the reviewers are generally convinced by their merits, and concerns about novelty have been addressed during the discussion. Another concern was the lack of evaluation on task-oriented dialog (TOD), but the preliminary results provided in the author response seems satisfactory, and I would encourage the authors to expand the preliminary TOD results in the paper. As these contributions seem sufficiently noteworthy and all reviewers agreed to accept the paper, I also give an "accept" recommendation.

**Note From Pc:**

if the above contains the word "oral" or "spotlight" please see: "oral" presentation means -> notable-top-5% and "spotlight" means -> notable-top-25%. As stated in our emails, we are disassociating presentation type from AC recommendations

**Summary Of Ac-Reviewer Meeting:**

The paper finally ended up in the non-borderline range (average score 6.5) and there wasn't much to discuss as all major concerns have been addressed and all reviewers gave recommendations above the acceptance threshold. The only serious concern (in the original reviews) was the novelty aspect raised by reviewer bCi9, but they finally agreed the approach is sufficiently novel from a Conversational AI standpoint, and this reviewer (as well as all other reviewers) are fine with "accept".